# TREE-TO-TREE NEURAL NETWORKS FOR PROGRAM TRANSLATION

**Xinyun Chen   Chang Liu   Dawn Song**
University of California, Berkeley

## ABSTRACT

Program translation is an important tool to migrate legacy code in one language into an ecosystem built in a different language. In this work, we are the first to consider employing deep neural networks toward tackling this problem. We observe that program translation is a modular procedure, in which a sub-tree of the source tree is translated into the corresponding target sub-tree at each step. To capture this intuition, we design a tree-to-tree neural network as an encoder-decoder architecture to translate a source tree into a target one. Meanwhile, we develop an attention mechanism for the tree-to-tree model, so that when the decoder expands one non-terminal in the target tree, the attention mechanism locates the corresponding sub-tree in the source tree to guide the expansion of the decoder. We evaluate the program translation capability of our tree-to-tree model against several state-of-the-art approaches. Compared against other neural translation models, we observe that our approach is consistently better than the baselines with a margin of up to 15 points. Further, our approach can improve the previous state-of-the-art program translation approaches by a margin of 20 points on the translation of real-world projects.

## 1 INTRODUCTION

Programs are the main tool for building computer applications, the IT industry, and the digital world. Various programming languages have been invented to facilitate programmers to develop programs for different applications. At the same time, the variety of different programming languages also introduces a burden when programmers want to combine programs written in different languages together. Therefore, there is a tremendous need to enable program translation between different programming languages.

Nowadays, to translate programs between different programming languages, typically programmers would manually investigate the correspondence between the grammars of the two languages, then develop a rule-based translator. However, this process can be inefficient and error-prone. In this work, we make the first attempt to examine whether we can leverage deep neural networks to build a program translator automatically.

Intuitively, the program translation problem in its format is similar to a natural language translation problem. Some previous work propose to adapt phrase-based statistical machine translation approaches (SMT) for code migration (Nguyen et al., 2013; Karaivanov et al., 2014; Nguyen et al., 2015). Recently, neural network approaches, such as sequence-to-sequence-based models (Bahdanau et al., 2015; Cho et al., 2015; Eriguchi et al., 2016; He et al., 2016; Vaswani et al., 2017), have achieved the state-of-the-art performance on machine translation. In this work, we study neural machine translation methods to handle the program translation problem. However, a big challenge making a sequence-to-sequence-based model ineffective is that, unlike natural languages, programming languages have rigorous grammars and are not tolerant to typos and grammatical mistakes. It has been demonstrated that it is very hard for an RNN-based sequence generator to generate syntactically correct programs when the lengths grow large (Karpathy et al., 2015).

In this work, we observe that the main issue of an RNN that makes it hard to produce syntactically correct programs is that it entangles two sub-tasks together: (1) learning the grammar; and (2) aligning the sequence with the grammar. When these two tasks can be handled separately, the performance can typically boost. For example, Dong & Lapata (2016) employs a tree-based decoder

to separate the two tasks. In particular, the decoder in (Dong & Lapata, 2016) leverages the tree structural information to (1) generate the nodes at the same depth of the parse tree using an LSTM decoder; and (2) expand a non-terminal and generate its children in the parse tree. Such an approach has been demonstrated to achieve the state-of-the-art results on several semantic parsing tasks.

Inspired by this observation, we hypothesize that the structural information in both input and output parse trees can be leveraged to enable such a separation. Inspired by this intuition, we propose tree-to-tree neural networks to combine both a tree encoder and a tree decoder. In particular, we observe that in the program translation problem, both input and output programs have their parse trees. In addition, a cross-language compiler typically follows a modular procedure to translate the individual sub-components in the source tree into their corresponding target ones, and then compose them to form the final target tree. Therefore, we design the workflow of a tree-to-tree neural network to align with this procedure: when the decoder expands a non-terminal, it locates the corresponding sub-tree in the source tree using an attention mechanism, and uses the information of the sub-tree to guide the non-terminal expansion. In particular, a tree encoder is helpful in this scenario, since it can aggregate all information of a sub-tree to the embedding of its root, so that the embedding can be used to guide the non-terminal expansion of the target tree.

We follow the above intuition to design the tree-to-tree translation model. Existing works (Socher et al., 2011b; Kusner et al., 2017) have proposed tree-based autoencoder architectures. However, in these models, the decoder can only access to a single hidden vector representing the source tree, thus they are not performant in the translation task. In contrast, we employ an attention mechanism for the decoder to access the source tree, so that our approach can significantly improve the translation performance (i.e., from $0\%$ to $> 90\%$). To the best of our knowledge, this is the first tree-to-tree neural network architecture proposed for translation tasks in the literature.

To test our hypothesis, we develop two novel program translation tasks, and employ a Java to C# benchmark used by existing program translation works (Nguyen et al., 2015; 2013). First, we compare our approach against several neural network approaches on our proposed two tasks. Experimental results demonstrate that our tree-to-tree model outperforms other state-of-the-art neural networks on the program translation tasks, and yields a margin of up to $5\%$ on the token accuracy and up to $15\%$ on the program accuracy. Further, we compare our approach with previous program translation approaches on the Java to C# benchmark, and the results show that our tree-to-tree model outperforms previous state-of-the-art by a large margin of $20\%$ on program accuracy. These results demonstrate that our tree-to-tree model is promising toward tackling the program translation problem. Meanwhile, we believe that our proposed tree-to-tree neural network could also be adapted to other tree-to-tree tasks, and we consider it as future work.

## 2 PROGRAM TRANSLATION PROBLEM

In this work, we consider the problem of translating a program in one language into another. One approach is to model the problem as a machine translation problem between two languages, and thus numerous neural machine translation approaches can be applied.

For the program translation problem, however, a unique property is that each input program unambiguously corresponds to a unique parse tree. Thus, rather than modeling the input program as a sequence of tokens, we can consider the problem as translating a source tree into a target tree. Note that most modern programming languages are accompanied with a well-developed parser, so we can assume that the parse trees of both the source and the target programs can be easily obtained.

The main challenge of the problem in our consideration is that the cross-compiler for translating programs from one language into another typically does not exist. Therefore, even if we assume the existence of parsers for both the source and the target languages, the translation problem itself is still non-trivial. We formally define the problem as follows.

**Definition 1** (Program translation). *Given two programming languages $\mathcal{L}_s$ and $\mathcal{L}_t$, each being a set of instances $(p_k, T_k)$, where $p_k$ is a program, and $T_k$ is its corresponding parse tree. We assume that there exists a translation oracle $\pi$, which maps instances in $\mathcal{L}_s$ to instances in $\mathcal{L}_t$. Given a dataset of instance pairs $(i_s, i_t)$ such that $i_s \in \mathcal{L}_s, i_t \in \mathcal{L}_t$ and $\pi(i_s) = i_t$, our problem is to learn a function $F$ that maps each $i_s \in \mathcal{L}_s$ into $i_t = \pi(i_s)$.*

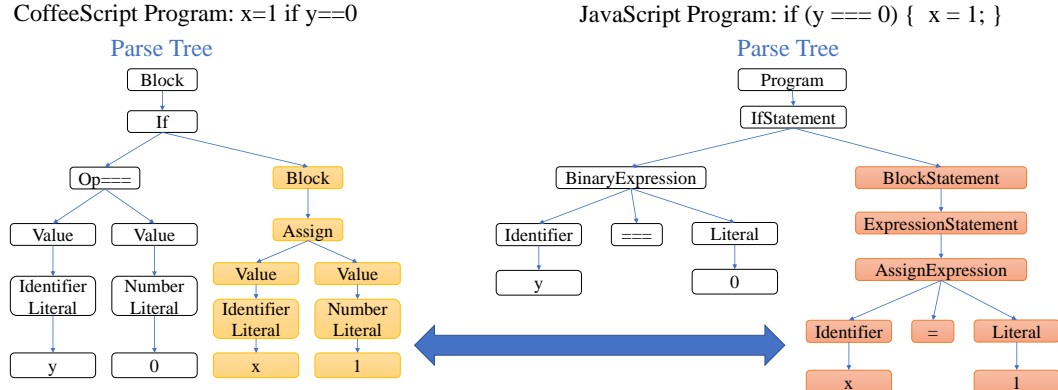

Figure 1: Translating a CoffeeScript program into JavaScript. The sub-component in the Coffee-Script program and its corresponding translation in JavaScript are highlighted.

Note that this problem definition does not restrict on whether $F$ knows only the program $p_k$ or only the parse tree $T_k$. Thus it can leverage both information to finish the translation task. Also, in the target language, since a program and its parse tree uniquely matches each other, $F$ only needs to predict one of them to be correct.

In this definition, $\mathcal{L}_s$ and $\mathcal{L}_t$ are the source and target languages respectively, and we use the oracle $\pi$ to model the goal of the translation. Notice that different tasks can be modeled with different choices of $\pi$. For example, besides program translation between two languages, we can also model the compiler optimization by setting $\mathcal{L}_s = \mathcal{L}_t$ and choosing $\pi$ that maps an un-optimized program into an optimized one.

In this work, we focus on the problem setting that we have a set of paired source and target programs to learn the translator. Note that all existing program translation works (Karaivanov et al., 2014; Nguyen et al., 2015; 2013) also study the problem under such an assumption. When such an alignment is lacking, the program translation problem is more challenging. Several techniques for NMT have been proposed to handle this issue, such as dual learning (He et al., 2016), which have the potential to be extended for the program translation task. We leave these more challenging problem setups as future work.

## 3 TREE-TO-TREE NEURAL NETWORK

In this section, we present our design of the tree-to-tree neural network. We first motivate the design, and then present the details.

### 3.1 PROGRAM TRANSLATION AS A TREE-TO-TREE TRANSLATION PROBLEM

Figure 1 presents an example of translation from CoffeeScript to JavaScript. We observe several interesting properties of the program translation problem. First, the translation can be modular. The figure highlights a sub-component in the source tree corresponding to x=1 and its translation in the target tree corresponding to x=1;. This correspondence is independent of other parts of the program. Consider when the program grows longer and this statement may repetitively occur multiple times, it may be hard for a sequence-to-sequence model to capture the correspondence based on only token sequences without structural information.

Second, such a correspondence makes it a natural solution to locate the referenced sub-tree in the source tree when expanding a non-terminal in the target tree into a sub-tree. Inspired by this intuition, we design the tree-to-tree neural network with attention, so that when decoding a non-terminal into a (sub-)tree, the model can employ an attention mechanism to locate the referenced source sub-tree. The soft-attention mechanism makes the model differentiable, so that it can be trained end-to-end.

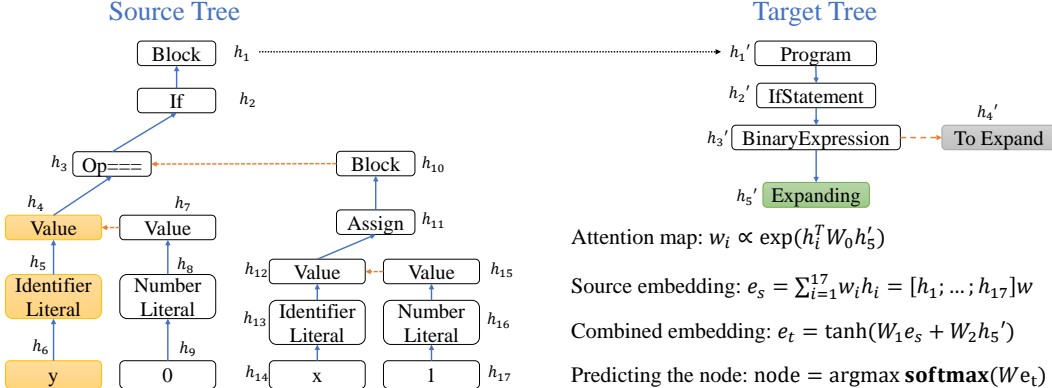

Figure 2: Tree-to-tree workflow: The arrows indicate the computation flow of the encoder-decoder architecture. Blue solid arrows indicate the flow from/to the left child, while orange dashed arrows are for the right child. The black dotted arrow from the source tree root to the target tree root indicates that the LSTM state is copied. The green box denotes the expanding node, and the grey one denotes the node to be expanded in a queue. The sub-tree of the source tree corresponding to the expanding node is highlighted in yellow. The right corner lists the formulas to predict the token for the expanding node.

Third, although the corresponding sub-trees are analogues to each other, they differ in two aspects. On the one hand, the non-terminal nodes are different. For example, the non-terminal `IdentifierLiteral` in CoffeeScript is called `Identifier` in JavaScript. Some non-terminals, such as `Value`, may not even have a correspondence. On the other hand, the topology structures of the corresponding sub-trees can be different. For example, the `Assign` node in the source tree has two children, while its correspondence in the target tree has three: an additional = is inserted in the middle. These differences can pose a challenge to build a program translation model.

## 3.2    TREE-TO-TREE NEURAL NETWORK

A tree-to-tree neural network follows an encoder-decoder framework to encode the source tree into an embedding, and decode the embedding into the target tree. To capture the intuition of the modular translation process, the decoder employs an (soft) attention mechanism to locate the corresponding source sub-tree when expanding the non-terminal. We illustrate the workflow of a tree-to-tree model in Figure 2. In the following, we present each component of the model.

**Converting a tree into a binary one.**   Note that the input and output trees may contain multiple branches. Although we can design tree-encoders and tree-decoders to handle trees with arbitrary number of branches, we observe that encoder and decoder for binary trees can be more effective. Thus, the first step is to convert both the source tree and the target tree into a binary tree. To this end, we employ the Left-Child Right-Sibling representation for this conversion.

**Binary tree encoder.**   The encoder employs a Tree-LSTM (Tai et al. (2015)) to compute embeddings for both the entire tree and each sub-tree. In particular, consider a node $N$ that is attached with a token $t_s$ in its one-hot encoding representation, and it has two children $N_L$ and $N_R$, which are its left child and right child respectively. The encoder recursively computes the embedding for $N$ from the bottom up.

Assume that the left child and the right child maintain the LSTM state $(h_L, c_L)$ and $(h_R, c_R)$ respectively. Then the LSTM state $(h, c)$ of $N$ is computed as

$$(h, c) = \text{LSTM}(([h_L; h_R], [c_L; c_R]), x) \tag{1}$$

where $[a; b]$ indicates the concatenation of two vectors $a$ and $b$. Note that a node may lack one or both of its children. In this case, the encoder sets the LSTM state of the missing child to be zero.

**Binary tree decoder.** The decoder generates the target tree starting from a single root node. The decoder first copies the LSTM state $(h, c)$ of the root of the source tree, and attaches it to the root node of the target tree. Then the decoder maintains a queue of all nodes to be expanded, and recursively expands each of them. In each iteration, the decoder pops one node from the queue, and expands it. In the following, we call the node being expanded *the expanding node*.

First, the decoder will predict the token of expanding node. To this end, the decoder computes the embedding $e_t$ of the expanding node $N$, and then feeds it into a softmax regression network for prediction:

$$t_t = \mathbf{argmax}\, \mathbf{softmax}(We_t) \tag{2}$$

Here, $W$ is a trainable matrix of size $V_t \times d$, where $V_t$ is the vocabulary size of the outputs and $d$ is the embedding dimension. Note that $e_t$ is computed using the attention mechanism, which we will explain later.

Each token $t_t$ is a non-terminal, a terminal, or a special $\langle \text{EOS} \rangle$ token. If $t_t = \langle \text{EOS} \rangle$, then the decoder finishes expanding this node. Otherwise, the decoder generates one new node as the left child and another new node as the right child of the expanding one. Assume that $(h', c')$, $(h'', c'')$ are the LSTM states of its left child and right child respectively, then they are computed as:

$$(h', c') = \text{LSTM}_L((h, c), Bt_t) \tag{3}$$

$$(h'', c'') = \text{LSTM}_R((h, c), Bt_t) \tag{4}$$

Here, $B$ is a trainable word embedding matrix of size $d \times V_t$. Note that the generation of the left child and right child use two different sets of parameters for $\text{LSTM}_L$ and $\text{LSTM}_R$ respectively. These new children are pushed into the queue of all nodes to be expanded.

The target tree generation process terminates when the queue is empty.

**Attention mechanism to locate the source sub-tree.** Now we consider how to compute $e_t$. One straightforward approach is to compute $e_t$ as $h$, which is the hidden state attached to the expanding node. However, in doing so, the embedding will soon forget the information about the source tree when generating deep nodes in the target tree, and thus the model yields a very poor performance.

To make better use of the information of the source tree, our tree-to-tree model employs an attention mechanism to locate the source sub-tree corresponding to the sub-tree rooted at the expanding node. Specifically, we compute the following probability:

$$P(N_s \text{ is the source sub-tree corresponding to } N_t | N_t)$$

where $N_t$ is the expanding node. We denote this probability as $P(N_s | N_t)$, and we compute it as

$$P(N_s | N_t) \propto \mathbf{exp}(h_s^T W_0 h_t) \tag{5}$$

where $W_0$ is a trainable matrix of size $d \times d$. To leverage the information from the source tree, the decoder can sample a source node $N_s$ following $P(N_s | N_t)$ or simply predict the most-likely source node $N_s = \mathbf{argmax}_{N_s} P(N_s | N_t)$, and then get its hidden $h_s$ as $e_s$. This embedding can then be combined with $h$, the hidden state of the expanding node, to compute $e_t$ as follows:

$$e_t = \mathbf{tanh}(W_1[h; e_s]) \tag{6}$$

where $W_1$ is a trainable matrix of size $d \times 2d$, and $[h; e_s]$ denotes the concatenation of $h$ and $e_s$.

However, this approach suffers a big issue that the entire network is no longer end-to-end differentiable unless explicit supervision is given to train $P(N_s | N_t)$. Unfortunately, this supervision is unavailable. We may rely on reinforcement learning algorithms to mitigate this issue, but doing so is expensive and may not be effective.

We solve this problem by changing the above hard-attention scheme to a soft-attention mechanism. Note that in Equation (6), we only need a continuous embedding vector $e_s$ rather than the discrete choice of $N_s$. Therefore, we can approximate $e_s$ as the expectation of the hidden state value across all $N_s$ conditioning on $N_t$. In fact, we have

$$e_s = \mathrm{E}[h_{N_s} | N_t] = \sum_{N_s} h_{N_s} \cdot P(N_s | N_t) \tag{7}$$

Note that this final formula (7) coincides with the standard attention formalism. Combining all equations from (1) to (7), the entire tree-to-tree neural network is fully differentiable and can be trained end-to-end.

```
Source program          Target program
for i=1; i<10; i+1 do   letrec f i =
    if x>1 then             if i<10 then
        y=1                     let _ = if x>1 then
    else                            let y=1 in ()
        y=2                         else let y=2 in ()
endfor                          in f i+1
                            else ()
                        in f 1
```

Figure 3: An example of the translation for the synthetic task.

**Parent attention feeding.** In the above approach, the attention vectors $e_t$ are computed independently to each other, since once $e_t$ is used for predicting the token $t_t$, $e_t$ is no longer used for further predictions. However, intuitively, the attention decisions for the prediction of each node should be related to each other. For example, for a non-terminal node $N_t$ in the target tree, suppose that it is related to $N_s$ in the source tree, then it is very likely that the attention weights of its children should focus on the descendants of $N_s$. Therefore, when predicting the attention vector of a node, the model should leverage the attention information of its parent as well.

Following this intuition, we propose a *parent attention feeding* mechanism, so that the attention vector of the expanding node is taken into account when predicting the attention vectors of its children. Formally, besides the embedding of the token $t_t$, we modify the inputs to $\text{LSTM}_L$ and $\text{LSTM}_R$ of the decoder in Equations (3) and (4) as below:

$$(h', c') = \text{LSTM}_L((h, c), [Bt_t; e_t]) \tag{8}$$

$$(h'', c'') = \text{LSTM}_R((h, c), [Bt_t; e_t]) \tag{9}$$

Notice that these formulas in their formats coincide with the input-feeding method for sequential neural networks (Luong et al., 2015), but their meanings are different. For sequential models, the input attention vector belongs to the previous token, while here it belongs to the parent node.

## 4 EVALUATION

In this section, we evaluate our tree-to-tree neural network with several baseline approaches on the program translation task. To do so, we first describe three benchmark datasets in Section 4.1 for evaluating different aspects. Then we evaluate our tree-to-tree model with several baseline approaches, including the state-of-the-art neural network approaches and program translation approaches. In the following, we start with presenting the details of the benchmark datasets and models, and then present the results.

### 4.1 DATASETS

To evaluate different approaches for the program translation problem, we employ three tasks: (1) a synthetic translation task from an imperative language to a functional language; (2) translation between CoffeeScript and JavaScript, which are both full-fledged languages; and (3) translation of real-world projects from Java to C#, which has been used as a benchmark in the literature.

For the synthetic task, we design an imperative source language and a functional target language. Such a design makes the source and target languages use different programming paradigms, so that the translation can be challenging. Figure 3 illustrates an example of the translation, which demonstrates that a for-loop is translated into a recursive function. We manually implement a translator, which is used to acquire the ground truth. The formal grammars of the two languages and the implementation of the translator can be found in Appendix D.1 and Appendix E respectively .

For the CoffeeScript-JavaScript task, the programming paradigms of the two languages are identical. CoffeeScript employs a Python-style succinct syntax, while JavaScript employs a C-style verbose

| Project | # of matched methods |
|---|---|
| Lucene (luc) | 5,516 |
| POI (poi) | 3,153 |
| Itext (ite) | 3,079 |
| JGit (jgi) | 2,780 |
| JTS (jts) | 2,003 |
| Antlr (ant) | 465 |
| Total | 16,996 |

Table 1: Statistics of the Java to C# dataset.

syntax. To control the program lengths of the training and test data, we develop a pCFG-based program generator and a subset of the core CoffeeScript grammar. We also limit the set of variables and literals to restrict the vocabulary size. We rely on the CoffeeScript compiler to generate the corresponding ground truth JavaScript programs. The grammar used to generate the programs in our experiments can be found in Appendix D.2 . In doing so, we can get a set of CoffeeScript-JavaScript pairs, and thus we can build a CoffeeScript-to-JavaScript dataset, and a JavaScript-to-CoffeeScript dataset by exchanging the source and the target.

For both of the above tasks, we randomly generate 100,000 pairs of source and target programs for training, 10,000 pairs as the development set, and 10,000 pairs for testing. More statistics of the above datasets can be found in Appendix B .

To build the Java to C# dataset, we employ the same approach as in (Nguyen et al., 2015) to crawl several open-source projects, which have both a Java and a C# implementation. Same as in (Nguyen et al., 2015), we pair the methods in Java and C# based on their file names and method names. The statistics of the dataset is summarized in Table 1. Due to the change of the versions of these projects, the concrete dataset in our evaluation may differ from (Nguyen et al., 2015). We apply ten-fold validation for matched method pairs in each project, as in (Nguyen et al., 2015).

## 4.2 METRICS

In our evaluation, we measure the following two metrics:

- **Token accuracy.** Given a set of source programs, we run the evaluated model to translate them into target programs, and calculate the percentage of the tokens that can match the ground truth.

- **Program accuracy.** Given a set of source programs, we run the evaluated model to translate them into target programs, and calculate the percentage of the predicted programs that can entirely match the ground truth.

Note that the program accuracy is an underestimation of the true accuracy based on semantic equivalence, and this metric has been used in (Nguyen et al., 2015). This metric is more meaningful than other previously proposed metrics, such as syntax-correctness and dependency-graph-accuracy, which are not directly comparable to semantic equivalence. In addition, token accuracy can provide more insight into the performance of different models.

## 4.3 MODEL DETAILS

We evaluate our tree-to-tree model against a sequence-to-sequence model (Bahdanau et al., 2015; Vinyals et al., 2015), a sequence-to-tree model (Dong & Lapata, 2016), and a tree-to-sequence model (Eriguchi et al., 2016). Note that for a sequence-to-sequence model, there can be four variants to handle different input-output formats. For example, given a program, we can simply tokenize it into a sequence of tokens. We call this format as *raw program*, denoted as P. We can also use the parser to parse the program into a parse tree, and then serialize the parse tree as a sequence of tokens. Our serialization of a tree follows its depth-first traversal order, which is the same as (Vinyals et al., 2015). We call this format as *parse tree*, denoted as T. For both input and output formats, we can choose either P or T. Similarly, for a sequence-to-tree model, we have two variants based on its input format being either P or T, and for a tree-to-sequence model, we have two variants based on its output

|  | Tree2tree | | | Seq2seq | | | | Seq2tree | | Tree2seq | |
|---|---|---|---|---|---|---|---|---|---|---|---|
|  | T→T | T→T (-PF) | T→T (-Attn) | P→P | P→T | T→P | T→T | P→T | T→T | T→P | T→T |
| Token accuracy | | | | | | | | | | | |
| SYN-S | **99.99%** | 99.95% | 55.60% | 99.75% | 99.59% | 99.90% | 99.73% | 99.70% | 99.51% | 99.88% | 99.82% |
| SYN-L | **99.60%** | 96.68% | 34.48% | 68.31% | 45.28% | 67.37% | 35.01% | 96.95% | 97.41% | 97.08% | 95.88% |
| Program accuracy | | | | | | | | | | | |
| SYN-S | **99.76%** | 98.61% | 0% | 97.92% | 97.35% | 98.38% | 98.18% | 96.14% | 98.01% | 98.51% | 98.36% |
| SYN-L | **97.50%** | 57.42% | 0% | 12.19% | 0% | 9.19% | 0% | 67.34% | 68.11% | 91.35% | 87.84% |

Table 2: Token accuracy and program accuracy of different approaches for the synthetic task.

|  | Tree2tree | | | Seq2seq | | | | Seq2tree | | Tree2seq | |
|---|---|---|---|---|---|---|---|---|---|---|---|
|  | T→T | T→T (-PF) | T→T (-Attn) | P→P | P→T | T→P | T→T | P→T | T→T | T→P | T→T |
| CoffeeScript to JavaScript translation | | | | | | | | | | | |
| CJ-AS | **99.57%** | 98.80% | 0.09% | 90.51% | 79.82% | 92.73% | 89.13% | 86.52% | 88.50% | 96.96% | 92.18% |
| CJ-BS | **99.75%** | 99.67% | 0% | 97.44% | 16.26% | 98.05% | 93.89% | 91.97% | 88.22% | 96.83% | 78.77% |
| CJ-AL | **97.15%** | 71.52% | 0% | 21.04% | 0% | 0% | 0% | 80.82% | 78.60% | 82.55% | 46.94% |
| CJ-BL | **95.60%** | 78.61% | 0% | 19.26% | 9.98% | 25.35% | 42.08% | 76.12% | 76.21% | 83.61% | 26.83% |
| JavaScript to CoffeeScript translation | | | | | | | | | | | |
| JC-AS | **87.75%** | 85.11% | 0.09% | 83.07% | 86.13% | 73.88% | 86.31% | 86.86% | 86.99% | 71.61% | 86.53% |
| JC-BS | **86.37%** | 80.35% | 0% | 80.49% | 85.94% | 69.77% | 85.28% | 85.06% | 84.25% | 66.82% | 85.31% |
| JC-AL | **78.59%** | 54.93% | 0% | 77.10% | 77.30% | 65.52% | 75.70% | 77.11% | 77.59% | 60.75% | 75.75% |
| JC-BL | **75.62%** | 44.40% | 0% | 73.14% | 73.96% | 61.92% | 74.51% | 74.34% | 71.56% | 57.09% | 73.86% |

Table 3: Program accuracy of different approaches for translation between CoffeeScript and JavaScript. The token accuracy can be found in Appendix C .

format being either P or T. Note that the sequence-to-tree model generates a tree as output, and thus requires its output format to be T (unserialized). For the same reason, the tree-to-sequence model requires its input format to be T (unserialized), and our tree-to-tree only has one form. Therefore, we have 9 different models in our evaluation.

The hyper-parameters used in different models can be found in Appendix A . The baseline models have employed their own input-feeding or parent-feeding method that is analogous to our parent attention feeding mechanism.

## 4.4 RESULTS ON THE SYNTHETIC TASK

We create two datasets for the synthetic task: one with average length 20 (*SYN-S*) and the other with average length 50 (*SYN-L*). Here, the length of a program indicates the number of tokens in the source program.

The results are presented in Table 2. We can observe that our tree2tree model outperforms all baseline models on both datasets and with both metrics. Especially, on the dataset with longer programs, the program accuracy significantly outperforms all seq2seq models by a large margin, i.e., $> 85\%$. Its margin over a seq2tree model can also reach around 30 points. These results demonstrate that tree2tree model is more capable of learning the program translation relationship.

Notice that increasing the length of the programs indeed makes the problem harder: the program accuracy drops for all models. However, we observe that the decrease of our tree2tree model on the program accuracy is only around 2 points, which is much smaller than the decrease of other models. This shows that the tree2tree model is significantly better than other baselines at handling longer inputs.

In addition, we perform ablation study to compare the full tree2tree model with (1) tree2tree without parent attention feeding (T→T (-PF)) and (2) tree2tree without attention (T→T (-Attn)). We observe that the full tree2tree model significantly outperforms the other alternatives. In particular, on SYN-L, the full tree2tree's program accuracy is $40$ points higher than the tree2tree model without parent attention feeding.

More importantly, we observe that the program accuracy of tree2tree model without the attention mechanism is always $0\%$. Note that such a model is similar to a tree-to-tree autoencoder architecture.

| | Tree2tree | J2C# | 1pSMT | mppSMT |
|---|---|---|---|---|
| | | Reported in (Nguyen et al., 2015) | | |
| Lucene | **72.8%** | 21.5% | 21.6% | 40.0% |
| POI | **72.2%** | 18.9% | 34.6% | 48.2% |
| Itext | **67.5%** | 25.1% | 24.4% | 40.6% |
| JGit | **68.7%** | 10.7% | 23.0% | 48.5% |
| JTS | **68.2%** | 11.7% | 18.5% | 26.3% |
| Antlr | 31.9% (**58.3%**) | 10.0% | 11.5% | 49.1% |

Table 4: Program accuracy on the Java to C# translation. In the parentheses, we put the program accuracy that can be achieved by increasing the training set.

This result shows that our novel architecture can significantly outperform previous tree-to-tree-like architectures on the program translation task.

## 4.5   RESULTS ON THE COFFEESCRIPT-JAVASCRIPT TASK

We now move on to the CoffeeScript-JavaScript task. We create several datasets named as XY-ZW: X and Y (C or J) indicate the source and target language respectively; Z (A or B) indicates the vocabulary; and W (S or L) indicates the program length. In particular, vocabulary A uses only $\{x,y\}$ as variable names and $\{0,1\}$ as literals; vocabulary B uses all alphabetical characters as variable names, and all single digits as literals. S indicates that the CoffeeScript programs contain 10 tokens on average; and L for 20.

The program accuracy results are presented in Table 3, and the token accuracy results can be found in Appendix A . Most of the observations from our synthetic task remain: our tree2tree model outperforms all baseline models; all models perform worse on longer inputs; both the attention and the parent attention feeding mechanisms boost the performance of our tree2tree model significantly.

In addition, we observe that for the translation from JavaScript to CoffeeScript, the improvements of the tree2tree model over the baselines are much smaller than for CoffeeScript to JavaScript translation. We attribute this to the fact that the target programs are much shorter. For example, for a CoffeeScript program with 20 tokens, its corresponding JavaScript program may contain more than 300 tokens. Thus, the model needs to predict much less tokens for a CoffeeScript program than a JavaScript program, so that even seq2seq models can achieve a reasonably good accuracy. However, still, we can observe that our tree2tree model outperforms all baselines.

## 4.6   RESULTS ON REAL-WORLD PROJECTS

We now compare our approach with three state-of-the-art program translation approaches, i.e., J2C# (jav), 1pSMT (Nguyen et al., 2013), and mppSMT (Nguyen et al., 2015), on the real-world benchmark from Java to C#. Here, J2C# is a rule-based system, 1pSMT directly applies the phrase-based SMT on sequential programs, and mppSMT is a multi-phase phrase-based SMT approach that leverages both the raw programs and their parse trees. Our approach handles the out-of-vocabulary problem by canonicalizing all variable names, literals and string constants. For example, the first variable in a program (and its subsequent references) is renamed as v1, the second as v2, etc. In this way, the dataset contains only a small set of variable names.

The results are summarized in Table 4. For previous approaches, we report the results from (Nguyen et al., 2015). We can observe that our tree2tree approach can significantly outperform the previous state-of-the-art on all projects except Antlr. The improvements range from $20.2\%$ to $41.9\%$.

On Antlr, the tree2tree model performs worse. We attribute this to the fact that Antlr contains too few data samples for training. We test our hypothesis by constructing another training and validation set from all other 5 projects, and test the model on the entire Antlr. We observe that the model can achieve a test accuracy of $58.3\%$, which is 9 points higher than the state-of-the-art. Therefore, we conclude that our approach can significantly outperform previous program translation approaches when there are sufficient training data.

## 5  RELATED WORK

**Statistical approaches for program translation.**  Some recent work have applied statistical machine translation techniques to program translation (Allamanis et al., 2017; Karaivanov et al., 2014; Nguyen et al., 2015; 2013; 2016). For example, several works propose to adapt phrase-based statistical machine translation models and leverage grammatical structures of programming languages for code migration (Karaivanov et al., 2014; Nguyen et al., 2015; 2013). Nguyen et al. (2016) proposes to use Word2Vec representation for APIs in libraries used in different programming languages, then learns a transformation matrix for API mapping. On the contrary, our work is the first to employ deep learning techniques for program translation.

**Neural networks with tree structures.**  In recent years, various neural networks with tree structures have been proposed to employ the structured information of the data (Dong & Lapata, 2016; Rabinovich et al., 2017; Parisotto et al., 2017; Yin & Neubig, 2017; Alvarez-Melis & Jaakkola, 2017; Tai et al., 2015; Zhu et al., 2015; Socher et al., 2011a; Eriguchi et al., 2016; Zhang et al., 2016; Socher et al., 2011b; Kusner et al., 2017). In these work, different tree-structured encoders are proposed for embedding the input data, and different tree-structured decoders are proposed for predicting the output trees. In particular, in (Socher et al., 2011b; Kusner et al., 2017), they propose tree-structured autoencoders to learn vector representations of trees, and show better performance on tree reconstruction and other tasks such as sentiment analysis. In this work, we are the first to design the tree-to-tree neural network for translation tasks.

**Neural networks for parsing.**  Other work study using neural networks to generate parse trees from input-output examples (Dong & Lapata, 2016; Vinyals et al., 2015; Aharoni & Goldberg, 2017; Rabinovich et al., 2017; Yin & Neubig, 2017; Alvarez-Melis & Jaakkola, 2017; Dyer et al., 2016; Chen et al., 2018; 2016). Dong & Lapata (2016) proposes a seq2tree model that allows the decoder RNN to generate the output tree recursively in a top-down fashion. This approach achieves the state-of-the-art results on several semantic parsing tasks. Some other work incorporate the knowledge of the grammar into the architecture design (Yin & Neubig, 2017; Rabinovich et al., 2017) to achieve better performance on specific tasks. However, these approaches are hard to generalize to other tasks. Again, none of them is designed for program translation or proposes a tree-to-tree architecture.

**Neural networks for code generation.**  A recent line of research study using neural networks for code generation (Balog et al., 2017; Devlin et al., 2017; Parisotto et al., 2017; Ling et al., 2016; Rabinovich et al., 2017; Yin & Neubig, 2017). In (Ling et al., 2016; Rabinovich et al., 2017; Yin & Neubig, 2017), they study generating code in a DSL from inputs in natural language or in another DSL. However, their designs require additional manual efforts to adapt to new DSLs in consideration. In our work, we consider the tree-to-tree model as a generic approach that can be applied to any grammar.

## 6  CONCLUSION AND FUTURE WORK

In this work, we are the first to consider neural network approaches for the program translation problem, and are the first to propose a tree-to-tree neural network that combines both a tree-RNN encoder and a tree-RNN decoder for translation tasks. Extensive evaluation demonstrates that our tree-to-tree neural network outperforms several state-of-the-art models, including neural networks for machine translation and statistical machine translation-based program translation approaches. This renders our tree-to-tree model as a promising tool toward tackling the program translation problem. In addition, we believe that our proposed tree-to-tree neural network has potential to generalize to other tree-to-tree tasks, and we consider it as future work.

At the same time, we observe many challenges on program translation that existing techniques are not capable of handling. For example, the models are hard to generalize to programs longer than the training ones; it is unclear how to handle an infinite vocabulary set that may be employed in real-world applications; further, the training requires a dataset of aligned input-output pairs, which may be lacking in practice. We consider all these problems as important future work in the research agenda toward solving the program translation problem.

ACKNOWLEDGEMENT

We thank anonymous reviewers for their valuable feedbacks. This material is in part based upon work supported by the National Science Foundation under Grant No. TWC-1409915, Berkeley DeepDrive, and DARPA STAC under Grant No. FA8750-15-2-0104. Any opinions, findings, and conclusions or recommendations expressed in this material are those of the author(s) and do not necessarily reflect the views of the National Science Foundation.

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

|  | Seq2seq | Seq2tree | Tree2seq | Tree2tree |
|---|---|---|---|---|
| Batch size | 100 | 20 | 100 | 100 |
| Number of RNN layers | 3 | 1 | 1 | 1 |
| Encoder RNN cell | LSTM | LSTM | Tree LSTM | Tree LSTM |
| Decoder RNN cell | LSTM | | | |
| Initial learning rate | 0.005 | | | |
| Learning rate decay schedule | Decay the learning rate by a factor of $0.8\times$ when the validation loss does not decrease for 500 mini-batches | | | |
| Hidden state size | 256 | | | |
| Embedding size | 256 | | | |
| Dropout rate | 0.5 | | | |
| Gradient clip threshold | 5.0 | | | |
| Weights initialization | Uniformly random from [-0.1, 0.1] | | | |

Table 5: Hyper-parameters chosen for each neural network model.

|  | CJ-(A/B)S | CJ-(A/B)L | SYN-S | SYN-L |
|---|---|---|---|---|
| Average input length (P) | 10 | 20 | 20 | 50 |
| Minimal output length (P) | 23 | 33 | 22 | 46 |
| Maximal output length (P) | 151 | 311 | 44 | 96 |
| Average output length (P) | 44 | 69 | 30 | 71 |
| Minimal input length (T) | 34 | 69 | 40 | 100 |
| Maximal input length (T) | 61 | 111 | 56 | 134 |
| Average input length (T) | 48 | 85 | 49 | 111 |
| Minimal output length (T) | 38 | 73 | 41 | 90 |
| Maximal output length (T) | 251 | 531 | 82 | 177 |
| Average output length (T) | 71 | 129 | 55 | 133 |

Table 6: Statistics of the datasets used for the synthetic task and the CoffeeScript-JavaScript task.

## A    HYPER-PARAMETERS OF NEURAL NETWORK MODELS

We present the hyper-parameters of different neural networks in Table 5. These hyper-parameters are chosen to achieve the best validation accuracy through a grid search.

## B    MORE STATISTICS OF OUR PROPOSED DATASETS

We present more detailed statistics of the datasets for the synthetic task and the CoffeeScript-JavaScript task in Table 6.

## C    MORE RESULTS ON COFFEESCRIPT-JAVASCRIPT TASK

Table 7 shows the token accuracy of different approaches for the translation between CoffeeScript and JavaScript.

## D    GRAMMARS FOR GENERATING THE PROGRAMS IN THE EVALUATION

### D.1    GRAMMARS FOR THE SYNTHETIC TASK

The grammar specifications of the source language (FOR language) and the target language (LAMBDA language) used in the synthetic task are provided in Figure 4 and Figure 5 respectively.

| | Tree2tree | | | Seq2seq | | | | Seq2tree | | Tree2seq | |
|---|---|---|---|---|---|---|---|---|---|---|---|
| | T→T | T→T (-PF) | T→T (-Attn) | P→P | P→T | T→P | T→T | P→T | T→T | T→P | T→T |
| CoffeeScript to JavaScript translation | | | | | | | | | | | |
| CJ-AS | **99.97%** | **99.97%** | 56.21% | 93.51% | 92.30% | 95.46% | 95.05% | 93.29% | 95.94% | 98.96% | 98.09% |
| CJ-BS | **99.98%** | **99.98%** | 47.54% | 99.08% | 87.51% | 99.11% | 96.14% | 98.31% | 98.09% | 99.27% | 98.10% |
| CJ-AL | **99.37%** | 98.16% | 32.99% | 85.84% | 25.65% | 19.13% | 36.18% | 95.64% | 94.74% | 94.18% | 84.71% |
| CJ-BL | **99.36%** | 99.27% | 31.80% | 80.22% | 63.49% | 87.27% | 79.85% | 94.09% | 94.64% | 93.85% | 78.07% |
| JavaScript to CoffeeScript translation | | | | | | | | | | | |
| JC-AS | **99.14%** | 98.81% | 65.42% | 88.44% | 96.27% | 88.46% | 98.34% | 98.20% | 99.06% | 86.93% | 98.36% |
| JC-BS | **98.84%** | 98.18% | 55.22% | 86.85% | 97.92% | 85.98% | 98.09% | 96.93% | **98.84%** | 84.81% | 97.94% |
| JC-AL | **96.95%** | 92.65% | 42.23% | 88.09% | 95.94% | 87.19% | 95.04% | 93.51% | 96.59% | 84.57% | 94.63% |
| JC-BL | **96.48%** | 92.49% | 39.89% | 87.31% | 94.12% | 85.70% | 96.24% | 94.79% | 96.33% | 83.03% | 94.68% |

Table 7: Token accuracy of different approaches for translation between CoffeeScript and JavaScript.

```
<Expr>      ::=   
              |   <Const>
              |   <Expr> + 
              |   <Expr> + <Const>
              |   <Expr> − 
              |   <Expr> − <Const>

 <Cmp>      ::=   <Expr> == <Expr>
              |   <Expr> > <Expr>
              |   <Expr> < <Expr>

<Assign>    ::=    = <Expr>

  <If>      ::=   if <Cmp> then <statement>
                  else <statement> endif

 <For>      ::=   for  = <Expr> ;
                  <Cmp> ; <Expr> do
                  <Statement> endfor

<Single>    ::=   <Assign> | <If> | <For>

  <Seq>     ::=   <Single> ; <Single>
              |   <Seq> ; <Single>

<Statement> ::=   <Seq> | <Single>
```

Figure 4: Grammar for the source language FOR in the synthetic task.

## D.2 GRAMMAR FOR THE COFFEESCRIPT-JAVASCRIPT TASK

The grammar used to generate the CoffeeScript-JavaScript dataset, which is a subset of the core CoffeeScript grammar, is provided in Figure 6.

## E PYTHON IMPLEMENTATION OF THE TRANSLATOR FOR THE SYNTHETIC TASK

The python source code to implement the translator from a FOR program to a LAMBDA program in the synthetic task is provided in Figure 7.

```
<Unit>  ::=  ()

 <App>  ::=   <Expr>
        |    <App> <Expr>

<Expr>  ::=  
        |    <Expr> + 
        |    <Expr> − 

 <Cmp>  ::=  <Expr> == <Expr>
        |    <Expr> > <Expr>
        |    <Expr> < <Expr>

<Term>  ::=  <LetTerm> | <Expr> | <Unit>
        |    <IfTerm> | <App>

<LetTerm>  ::=  let  = <Term> in <Term>
           |    letrec   = <Term>
                in <Term>

<IfTerm>  ::=  if <Cmp> then <Term>
               else <Term>
```

Figure 5: Grammar for the target language LAMBDA in the synthetic task.

```
       <Expr>  ::=  
               |    <Const>
               |    <Expr> + 
               |    <Expr> + <Const>
               |    <Expr> * 
               |    <Expr> * <Const>
               |    <Expr> == 
               |    <Expr> == <Const>
     <Simple>  ::=   = <Expr>
               |    <Expr>
    <IfShort>  ::=  <Simple> if <Expr>
               |    <IfShort> if <Expr>
 <WhileShort>  ::=  <Simple> while <Expr>
               |    <WhileShort> while <Expr>
<ShortStatement>  ::=  <Simple> | <IfShort> | <WhileShort>
    <Statement>  ::=  <ShortStatement>
               |    if <Expr> 
 <indent+> <Block> <indent->
               |    while <Expr> 
 <indent+> <Block> <indent->
               |    if <Expr> 
 <indent+> <Block> <indent-> 
                    else 
 <indent+> <Block> <indent->
               |    if <Expr> then <ShortStatement> else <ShortStatement>
        <Block>  ::=  <Statement>
               |    <Block> 
 <Statement>
```

Figure 6: A subset of the CoffeeScript grammar used to generate the CoffeeScript-JavaScript dataset. Here, 
 denotes the newline character.

```python
def translate_from_for(self, ast):
    if type(ast) == type([]):
        if ast[0] == '<SEQ>':
            t1 = self.translate_from_for(ast[1])
            t2 = self.translate_from_for(ast[2])
            if t1[0] == '<LET>' and t1[-1] == '<UNIT>':
                t1[-1] = t2
                return t1
            else:
                return ['<LET>', 'blank', t1, t2]
        elif ast[0] == '<IF>':
            cmp = ast[1]
            t1 = self.translate_from_for(ast[2])
            t2 = self.translate_from_for(ast[3])
            return ['<IF>', cmp, t1, t2]
        elif ast[0] == '<FOR>':
            var = self.translate_from_for(ast[1])
            init = self.translate_from_for(ast[2])
            cmp = self.translate_from_for(ast[3])
            inc = self.translate_from_for(ast[4])
            body = self.translate_from_for(ast[5])
            tb = ['<LET>', 'blank', body, ['<APP>', 'func', inc]]
            func_body = ['<IF>', cmp, tb, '<UNIT>']
            translate = ['<LETREC>', 'func', var, func_body, ['<APP>', 'func', inc]]
            return translate
        elif ast[0] == '<ASSIGN>':
            return ['<LET>', ast[1], ast[2], '<UNIT>']
        elif ast[0] == '<Expr>':
            return ast
        elif ast[0] == '<Op+>':
            return ast
        elif ast[0] == '<Op->':
            return ast
        elif ast[0] == '<CMP>':
            return ast
    else:
        return ast
```

Figure 7: The Python code to translate a FOR program into a LAMBDA program.

