# OpenReview forum: "Tree-to-tree Neural Networks for Program Translation"
_ICLR.cc/2018/Conference — Invite to Workshop Track_

### Official Review · AnonReviewer3 · 2017-11-25

**Rating:** 6
**Confidence:** 4

**Review:**

This paper presents a tree-to-tree neural network for translating programs
written in one Programming language to another. The model uses soft attention
mechanism to locate relevant sub-trees in the source program tree when
decoding to generate the desired target program tree. The model is evaluated
on two sets of datasets and the tree-to-tree model outperforms seq2tree and
seq2seq models significantly for the program translation problem.

This paper is the first to suggest the tree-to-tree network and an interesting
application of the network for the program translation problem. The evaluation
results demonstrate the benefits of having both tree-based encoder and decoder.
The tree encoder, however, is based on the standard Tree-LSTM and the application
in this case is synthetic as the datasets are generated using a manual rule-based
translation.

Questions/Comments for authors:

The current examples are generated using a manually developed rule-based system.
As the authors also mention it might be challenging to obtain the aligned examples
for training the model in practice. What is the intended use case then of
training the model when the perfect rule-based system is already available?

How complex are the rules for translating the programs for the two datasets and what
type of logic is needed to write such rules? It would be great if the authors can
provide the rules used to generate the dataset to better understand the complexity
of the translation task.

There are several important details missing regarding the baselines. For the
seq2seq and seq2tree baseline models, are bidirectional LSTMs used for the encoder?
What type of attention mechanisms are used? Are the hyper-parameters presented in
Table 1 based on best training performance?

In section 4.3, it is mentioned that the current models are trained and tested on
programs of length 20 and 50. Does the dataset contain programs of length upto
20/50 or exactly of length 20/50? How is program length defined -- in terms of
tree nodes or the number of lines in the program?

What happens if the models trained with programs upto length 20 are evaluated on
programs of larger length say 40? It would be interesting to observe the
generalization capabilities of all the different models.

There are two benefits of using the tree2tree model: i) use the grammar of the
language, and ii) use the structure of the tree for locating relevant sub-trees
(using attention). From the current evaluation results, the empirical benefit
of using the attention is not clear. How would the accuracies look when using
the tree2tree model without attention or when attention vector e_t is set to the
hidden state h of the expanding node?

---

> ### Author Response · Authors · 2017-12-25
> **Response**
>
> Thank you for your valuable comments! We respond to the questions below.
>
> The reviewer asks about the meaning of the two program translation tasks studied in our work. We have explained that this is a first step to understand the problem of using a deep neural network approach to solve the program translation problem, and we consider the more challenging task without aligned input-output pairs as an important future direction. Also, although we do not evaluate it in our work, we believe the study of tree-to-tree translation model may have applications to other tree-to-tree translation tasks.
>
> The CoffeeScript-to-JavaScript compiler is available online, which is too complex to explain in the paper. We have added the code of our translator between two synthetic languages in the appendix.
>
> To clarify some implementation details, our seq2seq model and seq2tree model faithfully implement [1] and [2]. In particular, we only use uni-directional LSTMs for the encoder, and the attention mechanism is the same as described in the original papers as well. We did grid search for hyper-parameters, and chose the best one based on their performance on the validation set.
>
> A program’s length is defined to be the total number of tokens in the program, and is guaranteed to be equal to 20/50.
>
> When we train the model on shorter programs (e.g., programs of length 20), then evaluate on longer programs (e.g., programs of length 50), the test accuracy is 0 for all models, including our proposed tree-to-tree model and the baseline models. We consider solving the generalization issue as the next important problem that we want to address in the future.
>
> We have clarified all above details in our revised version as well.
>
> We are running more experiments to provide a complete ablation study to understand the effectiveness of attention. In some preliminary results, we observe that the performance decreases dramatically when attention is not used. We will update the results once we finish the experiments.
>
> [1] Oriol Vinyals, Lukasz Kaiser, Terry Koo, Slav Petrov, Ilya Sutskever, Geoffrey Hinton. Grammar as a foreign language. NIPS 2015.
> [2] Li Dong and Mirella Lapata. Language to logical form with neural attention. ACL 2016.

---

> > ### Author Response · Authors · 2018-01-05
> > **We have added results of our model without the attention mechanism**
> >
> > We have updated the revision to include results of our tree2tree model without the attention mechanism, and observe that the performance decreases significantly. In particular, the program accuracy drops to nearly 0%. More details can be found in the paper.

---

### Official Review · AnonReviewer1 · 2017-11-27
**Some interesting components, but overstates contribution**

**Rating:** 4
**Confidence:** 3

**Review:**

This paper aims to translate source code from one programming language to another using
a neural network architecture that maps trees to trees. The encoder uses an upward pass of
a Tree LSTM to compute embeddings for each subtree of the input, and then the decoder
constructs a tree top-down. As nodes are created in the decoder, a hidden state is passed
from parents to children via an LSTM (one for left children, one for right children), and
an attention mechanism allows nodes in the decoder to attend to subtrees in the encoder.

Experimentally, the model is applied to two synthetic datasets, where programs in the
source domain are sampled from a PCFG and then translated to the target domain with a
hand-coded translator. The model is then trained on these pairs. Results show that the
nproposed approach outperforms sequence representations or serialized tree representations
of inputs and outputs.

Pros:

- Nice model which seems to perform well.

- Reasonably clear explanation.

A couple questions about the model:

- the encoder uses only bottom-up information to determine embeddings of subtrees. I wonder
if top-down information would create embeddings with more useful information for the attention
in the decoder to pick up on.

- I would be interested to know more details about how the hand-coded translator works. Does
it work in a context-free, bottom-up fashion? That is, recursively translate two children nodes
and then compute the translation of the parent as a function of the parent node and
translations of the two children? If so, I wonder what is missing from the proposed model
that makes it unable to perfectly solve the first task?

Cons:

- Only evaluated on synthetic programs, and PCFGs are known to generate unrealistic programs,
so we can only draw limited conclusions from the results.

- The paper overstates its novelty and doesn't properly deal with related work (see below)

The paper overstates its novelty and has done a poor job researching related work.
Statements like "We are the first to consider employing neural network approaches
towards tackling the problem [of translating between programming languages]" are
obviously not true (surely many people have *considered* it), and they're particularly
grating when the treatment of related  work is poor, as it is in this paper. For example,
there are several papers that frame the code migration problem as one of statistical
machine translation (see Sec 4.4 of [1] for a review and citations), but this paper
makes no reference to them. Further, [2] uses distributed representations for the purpose
of code migration, which I would call a "neural network approach," so there's not any
sense that I can see in which this statement is true. The paper further says, "To the best
of our knowledge, this is the first tree-to-tree neural network architecture in the
literature." This is worded better, but it's definitely not the first tree-to-tree
neural network. See, e.g., [3, 4, 5], one of which is cited, so I'm confused about
this claim.

In total, the model seems clean and somewhat novel, but it has only been tested on
unrealistic synthetic data, the framing with respect to related work is poor, and the
contributions are overstated.


[1] https://arxiv.org/abs/1709.06182
[2]  Trong Duc Nguyen, Anh Tuan Nguyen, and Tien N Nguyen. 2016b. Mapping API elements for code migration with
vector representations. In Proceedings of the International Conference on Software Engineering (ICSE).
[3] Socher, Richard, et al. "Semi-supervised recursive autoencoders for predicting sentiment distributions." Proceedings of the conference on empirical methods in natural language processing. Association for Computational Linguistics, 2011.
[4] https://arxiv.org/abs/1703.01925
[5] Parisotto, Emilio, et al. "Neuro-symbolic program synthesis." arXiv preprint arXiv:1611.01855 (2016).

---

> ### Author Response · Authors · 2017-12-25
> **Response and some clarifications**
>
> Thank you for the valuable comments!
>
> Thank you for pointing out these related work! We have revised our paper to carefully compare with more prior work. From a high level, the papers cited in Section 4.4 of [1] are not neural network models; [2] uses word2vec, which is simply to learn a lookup table. Therefore, we also do not consider [2] as a deep neural network approach; [3, 4] propose tree-structured autoencoder, which is a generative model, rather than a translation model. The key difference is that a translation model has access to the source tree, while a generative model does not. Therefore, we think it is fair to claim our work as “the first deep neural network approach for the tree-to-tree translation problem.”
>
> In addition, the reviewer mentions [5] as a tree-to-tree model, which is definitely not true. In fact, [5] is a sequence-to-tree model: the input of the model proposed in [5] is a sequence rather than a tree.
>
> We clarify the questions below.
>
> We use the bottom-up fashion to aggregate the information so that each tree node contains all information of its descendants. Propagating information from top to bottom does not match our intuition that the attention is allocated based on the sub-trees of the source tree.
>
> The hand-coded translator is in a bottom-up fashion, but not context-free. To construct some parents, the translator may need to manipulate its two children. We have added the code of our translator for the synthetic task in the appendix.

---

### Official Review · AnonReviewer2 · 2017-11-27
**New but trivial model, poor experiments**

**Rating:** 4
**Confidence:** 4

**Review:**

Authors proposed a neural network based machine translation method between two programming languages. The model is based on both source/target syntax trees and performs an attentional encoder-decoder style network over the tree structure.

The new things in the paper are the task definition and using the tree-style network in both encoder and decoder. Although each structure of encoder/decoder/attention network is based on the application of some well-known components, unfortunately, the paper pays much space to describe them. On the other hand, the whole model structure looks to be easily generalized to other tree-to-tree tasks and might have some potential to contribute this kind of problems.

In experimental settings, there are many shortages of the description. First, it is unclear that what the linearization method of the syntax tree is, which could affect the final model accuracy. Second, it is also unclear what the method to generate train/dev/test data is. Are those generated completely randomly? If so, there could be many meaningless (e.g., inexecutable) programs in each dataset. What is the reasonableness of training such kind of data, or are they already avoided from the data? Third, the evaluation metrics "token/program accuracy" looks insufficient about measuring the correctness of the program because it has sensitivity about meaningless differences between identifier names and some local coding styles.

Authors also said that CoffeeScript has a succinct syntax and Javascript has a verbose one without any agreement about what the syntax complexity is. Since any CoffeeScript programs can be compiled into the corresponding Javascript programs, we should assume that CoffeeScript is the only subset of Javascript (without physical difference of syntax), and this translation task may never capture the whole tendency of Javascript. In addition, authors had generated the source CoffeeScript codes, which seems that this task is only one of "synthetic" task and no longer capture any real world's programs.
If authors were interested in the tendency of real program translation task, they should arrange the experiment by collecting parallel corpora between some unrelated programming languages using resources in the real world.

Global attention mechanism looks somewhat not suitable for this task. Probably we can suppress the range of each attention by introducing some prior knowledge about syntax trees (e.g., only paying attention to the descendants in a specific subtree).

Suggestion:
After capturing the motivation of the task, I suspect that the traditional tree-to-tree (also X-to-tree) "statistical" machine translation methods still can also work correctly in this task. The traditional methods are basically based on the rule matching, which constructs a target tree by selecting source/target subtree pairs and arranging them according to the actual connections between each subtree in the source tree. This behavior might be suitable to transform syntax trees while keeping their whole structure, and also be able to treat the OOV (e.g., identifier names) problem by a trivial modification. Although it is not necessary, it would like to apply those methods to this task as another baseline if authors are interested in.

---

> ### Author Response · Authors · 2017-12-25
> **Response and some clarifications**
>
> Thank you for your valuable comments! We clarify some confusions below, and we would greatly appreciate it if the reviewer could provide more feedbacks based on our response.
>
> We have updated our paper to provide more details about our experimental setup. We employ the S-expression to serialize the tree. For example, the parse tree of source program in Figure 1 (i.e., the parse tree of x = 1 if y == 0)
> is represented as
>
> (Block(If(Op===(Value(Identifier Literal(y))Value(Number Literal(0)))Block(Assign(Value(Identifier Literal(x))Value(Number Literal(1))))))
>
> This is the de facto standard approach used in the literature such as [1] and [2]. To the best of our knowledge, we are not aware of more effective ways to encode a tree. We would greatly appreciate it if the reviewer could provide alternatives that have been examined in the literature, and we would be happy to try them out.
>
> As described in Section 4.1, we use a pCFG to generate train/dev/test programs, while guaranteeing their lengths are equal to the value we specify. We think testing on randomly generated cases can effectively examine the correctness of the learned translator. Different from natural language translation, program translation task requires to handle all corner cases that may not be frequently seen in practice. Thus, using random test cases can effectively reach to all such corner cases, and we cannot agree with the reviewer that doing so is meaningless.
>
> Our program accuracy is an under-approximation of semantic equivalence, while token accuracy can provide a detailed measurement to understand an approach when the program accuracy is low. In this sense, these two metrics can capture some meaningful information about approaches’ effectiveness. Note that verifying if two programs are semantic-equivalent definition is a turing-complete problem, thus all metrics have to be an approximation to some degree. We consider proposing a better metric as future work.
>
> The reviewer comments on the syntax of CoffeeScript and JavaScript, and argues that CoffeeScript is a subset of JavaScript, on which we do not agree. The syntactical grammars of two languages do not imply their complexity class. Both of these two languages are Turing-complete, meaning any program in one language has a correspondence in another. Also, for both comments on the syntax and the complexity class, we do not see the direct implication on the later comments on that our synthetic task does not capture the real world programs. For the later comment, as we have explained above, our task is designed to capture different corner cases of a program translator, and we consider handling longer real-world examples as an important future direction.
>
> We are happy to try some traditional statistical machine translation baselines. Thanks for the suggestion!
>
> [1] Oriol Vinyals, Lukasz Kaiser, Terry Koo, Slav Petrov, Ilya Sutskever, Geoffrey Hinton. Grammar as a foreign language. NIPS 2015.
> [2] Li Dong and Mirella Lapata. Language to logical form with neural attention. ACL 2016.

---

> > ### Comment · AnonReviewer2 · 2018-01-11
> > **Response to Authors**
> >
> > - Linearization
> > Make sense, and I recommend to add at least 1 conversion example to the paper to guarantee reproducibility, because some accidental style errors in linearized texts may affect the results.
> >
> > - "Meaningless" test sets
> > The point of my concern is the problem which possibly not-few auto-generated codes have (not "all" of them). CFG rules basically represent a wider language than actual specification of "executable" codes, and auto-generating codes based on only CFG rules may include actually incorrect ones (e.g., the use-before-define error can occur using the BNFs in Appendix A, but this is one of the critical problems in many programming languages).
> > Considering about the case that there are non-ignorable amount of incorrect codes in the evaluation data, it becomes hard to declare the expressiveness of the proposed model in real problems too.
> >
> > - CoffeeScript and Javascript
> > Complexity class is not the point. CoffeeScript has an explicit map (compiler) to Javascript which is guaranteed by the concept of itself, so converting CoffeeScript to Javascript is too trivial to generalize the effectiveness of the program translation models for other languages, which do not have explicit maps between each other (even when they share some programming paradigms, such as Java and Python). I think that, the translation task in opposite side, i.e., translating "raw" Javascript codes (which are gathered or generated from scratch, not generated from CoffeeScript) to CoffeeScript, was more effective to discuss about this point.

---

### Author Response · Authors · 2018-01-05
**A revision of the paper has been uploaded**

We have updated the paper with the following changes:

(1) We include more discussion of related work in Section 5, especially addressing the relationship between our work with previous program translation work using statistical machine translation methods, and with tree-structured autoencoder work.
(2) We provide more details about our experimental setup in Section 4, and include the implementation of the translator between two synthetic languages in the appendix.
(3) We have included results of our tree2tree model without the attention mechanism, and observe that the performance degrades dramatically.

---

### Decision · Program_Chairs · 2018-01-29
**ICLR 2018 Conference Acceptance Decision**

**Decision:**

Invite to Workshop Track

**Comment:**

the problem is interesting, and the approach is also interesting. however, the reviewers have found that this manuscript would benefit from more experiments, potentially involving some real data (even at least for evaluation) in addition to largely synthetic data sets used in the submission. i also agree with them and encourage authors to consider this option.